# Synthesis and Anti-Angiogenic Activity of Novel c(RGDyK) Peptide-Based JH-VII-139-1 Conjugates

**DOI:** 10.3390/pharmaceutics15020381

**Published:** 2023-01-22

**Authors:** George Leonidis, Anastasia Koukiali, Ioanna Sigala, Katerina Tsimaratou, Dimitris Beis, Thomas Giannakouros, Eleni Nikolakaki, Vasiliki Sarli

**Affiliations:** 1Department of Chemistry, Aristotle University of Thessaloniki, University Campus, 54124 Thessaloniki, Greece; 2Zebrafish Disease Model Laboratory, Biomedical Research Foundation Academy of Athens, 11527 Athens, Greece

**Keywords:** cancer, angiogenesis, peptide drug conjugates, PDCs, hybrid molecules, RGD peptide, receptor-mediated endocytosis, integrins, SRPK1

## Abstract

Peptide–drug conjugates are delivery systems for selective delivery of cytotoxic agents to target cancer cells. In this work, the optimized synthesis of JH-VII-139-1 and its c(RGDyK) peptide conjugates is presented. The low nanomolar SRPK1 inhibitor, JH-VII-139-1, which is an analogue of Alectinib, was linked to the α_ν_β_3_ targeting oligopeptide c(RGDyK) through amide, carbamate and urea linkers. The chemostability, cytotoxic and antiangiogenic properties of the synthesized hybrids were thoroughly studied. All conjugates retained mid nanomolar-level inhibitory activity against SRPK1 kinase and two out of four conjugates, **geo75** and **geo77** exhibited antiproliferative effects with low micromolar IC_50_ values against HeLa, K562, MDA-MB231 and MCF7 cancer cells. The activities were strongly related to the stability of the linkers and the release of JH-VII-139-1. In vivo zebrafish screening assays demonstrated the ability of the synthesized conjugates to inhibit the length or width of intersegmental vessels (ISVs). Flow cytometry experiments were used to test the cellular uptake of a fluorescein tagged hybrid in MCF7 and MDA-MB231 cells that revealed a receptor-mediated endocytosis process. In conclusion, most conjugates retained the inhibitory potency against SRPK1 as JH-VII-139-1 and demonstrated antiproliferative and antiangiogenic activities. Further animal model experiments are needed to uncover the full potential of such peptide conjugates in cancer therapy and angiogenesis-related diseases.

## 1. Introduction

Angiogenesis is described as the formation of new blood vessels from pre-existing vessels [1,2]. It is a vital physiological process for growth, development, and wound repair by which various materials, oxygen and nutrients are delivered to tissues and cellular wastes are removed. In cancer, abnormal angiogenesis is related to cell growth, tumorigenesis, and metastases [3,4,5]. However, recent findings implicate angiogenesis in many other diseases [6], such as diabetic retinopathy, neovascular glaucoma [7], autoimmune diseases, multiple sclerosis [8], rheumatoid arthritis [9], cardiovascular diseases [10], atherosclerosis [11], and cerebral ischemia. Angiogenesis is regulated by many pathways and signaling molecules such as the vascular endothelial growth factor (VEGF), the transforming growth factor (TGF)-β, the angiopoietin-1 and 2, the placental growth factor (PGF), α_v_β_3_ and α_v_β_5_ integrins [12], the Notch, Akt, Jak/STAT, ephrin/Eph signaling pathways and others [13,14]. VEGF binds to receptor VEGFR and this interaction predominantly regulates pathological and physiological angiogenesis [15,16,17]. The VEGFR signaling is monitored at multiple levels [18]. For example, serine-arginine (SR) protein kinases (SRPKs) regulate the splicing of proangiogenic VEGF165 by phosphorylation of the serine/arginine splicing factor 1 (SRSF1) [19]. Therefore, targeting SRPK1 kinase to regulate angiogenesis in angiogenic disorders has been proposed by several research groups. Bates et al. has used the SRPK1 inhibitor SRPIN340 to reduce proangiogenic VEGF165 and inhibit melanoma tumor growth in vivo [20]. Oltean et al. demonstrated that SPHINX and SRPIN340 treatment changed the expression of VEGF165 towards the anti-angiogenic splice isoform VEGF165b and decreased tumor growth in orthotopic PC3 models in mice [21].

In previous research efforts by our group, the cyclic pentapeptide c(RGDyK), which is a potent integrin α_v_β_3_ ligand, was employed for the targeted delivery of various anticancer drugs and bioactive molecules such as gemcitabine [22], the triterpenoids cucurbitacins [23], and platinum complexes [24]. c(RGDyK) peptide potently binds α_v_β_3_ integrins (IC_50_ = 3.8 ± 0.42 nM) and, less potently, α_v_β_5_ (IC_50_ = 503 ± 55 nM), α_v_β_6_ (IC_50_ = 86 ± 7 nM), and α_5_β_1_ integrins (IC_50_ = 236 ± 45 nM) [25]. In earlier studies, the SRPIN803 inhibitor which binds to SRPK1 and CK2 kinases was coupled to c(RGDyK) peptide aiming to form a new conjugate with improved antiangiogenic activities compared to both the peptide and the SRPIN803 (Figure 1) [26]. Although the antiangiogenic activities of the synthesized SRPIN803-c(RGDyK) hybrids and SRPIN803 were evident in zebrafish embryos, an unfavorable stability profile of SRPIN803 was observed as SRPIN803 and its derivatives undergo a retro-Knoevenagel reaction. As a next step, we intended to develop new anti-angiogenic compounds based on the more potent SRPK1 inhibitor, JH-VII-139-1 [27]. JH-VII-139-1 came up from the optimization of FDA-approved ALK inhibitor Alectinib [28], which previously was shown to bind to SRPK1. Gray and coworkers demonstrated that JH-VII-139-1 potently inhibits SRPK1 with an IC_50_ value of 1.1 nM and blocks angiogenesis in an age-related macular degeneration (AMD) animal model [27]. In this work, the synthesis of JH-VII-139-1 peptide conjugates with c(RGDyK) and the properties of the new hybrid compounds against SRPK1, cancer cell growth and angiogenesis are reported.

## 2. Materials and Methods

### 2.1. Chemistry

#### 2.1.1. General Experimental Details

All reactions were carried out under an atmosphere of Argon unless otherwise specified. Commercial reagents were used without further purification. Reactions were monitored by TLC and using UV light as a visualizing agent and aqueous ceric sulfate/phosphomolybdic acid, ethanolic p-anisaldehyde solution, potassium permanganate solution, and heat as developing agents. The ^1^H and ^13^C NMR spectra were recorded at 500 and 126 MHz (Agilent) with tetramethylsilane as an internal standard. Chemical shifts are indicated in δ values (ppm) from internal reference peaks (TMS ^1^H 0.00; CDCl_3_ ^1^H 7.26, ^13^C 77.16, (CD_3_)_2_SO ^1^H 2.50, ^13^C 39.52, (CD_3_)_2_CO ^1^H 2.05, ^13^C 29.84, 206.26). LC-MS analysis was performed on a LC-20AD Shimadzu connected to Shimadzu LCMS-2010EV (Shimadzu Kyoto, Japan) equipped with C18 analytical column (Supelco discovery C18, 5 μm 250 × 4.6 mm).

#### 2.1.2. Synthetic Procedures

7-Methoxy-1,1-dimethyl-3,4-dihydronaphthalen-2(1*H*)-one, **2**. To a solution of 7-methoxy-2-tetralone (5 g, 28.5 mmol) in THF (20 mL) were added tetrabutylammonium bisulfate (1.53 g, 4.5 mmol), methyl iodide (5.25 mL, 85.5 mmol), and 50% aqueous KOH solution (32.5 mL) and stirred at room temperature for 2 h. The mixture was diluted with EA and washed with water and brine. The organic layer was dried over Na_2_SO_4_ and concentrated under reduced pressure. The residue was purified by flash column chromatography (gradient elution PS–EA, 15:1 to 10:1) to afford the desired product, **2** (5.3 g, 91%) as a white solid. The spectral data were in accordance with those reported in the literature [29]. **2**: ^1^H NMR (500 MHz, CDCl_3_) δ 7.09 (d, *J* = 8.3 Hz, 1H), 6.88 (d, *J* = 2.5 Hz, 1H), 6.74 (dd, *J* = 8.3, 2.5 Hz, 1H), 3.81 (s, 3H), 3.03 (t, *J* = 6.9 Hz, 2H), 2.66 (t, *J* = 6.9 Hz, 2H), 1.42 (s, 6H). ^13^C NMR (126 MHz, CDCl_3_) δ 214.7, 158.8, 145.0, 129.1, 127.5, 112.3, 111.4, 55.4, 48.0, 37.6, 27.8, 26.9. ESI-MS, positive mode: *m/z* calcd mass for C_13_H_16_O_2_Na [M + Na] ^+^ = 227.10, was found 227.05.

6-Bromo-7-methoxy-1,1-dimethyl-3,4-dihydronaphthalen-2(1*H*)-one, **3**. To a solution of **2** (2.00 g, 9.8 mmol) in CH_3_CN (40 mL) was added *N*-bromosuccinimide (1.92 g, 10.8 mmol). After stirring for 2.5 h at room temperature, the reaction mixture was concentrated under reduced pressure. The residue was purified by column chromatography (gradient elution PS–EA, 15:1 to 13:1) to yield **3** (2.55 g, 92%) as a white solid. ^1^H-NMR data were in accordance with those reported in the literature [30]. **3**: ^1^H NMR (500 MHz, CDCl_3_) δ 7.35 (s, 1H), 6.83 (s, 1H), 3.90 (s, 3H), 3.01 (t, *J* = 6.9 Hz, 2H), 2.65 (t, *J* = 6.9 Hz, 2H), 1.43 (s, 6H). ^13^C NMR (126 MHz, CDCl_3_) δ 213.8, 155.1, 144.2, 132.7, 129.0, 110.0, 109.7, 56.5, 48.0, 37.2, 27.5, 26.9.

3-Hydrazinylbenzonitrile, **5**. 4-Cyanoaniline (472 mg, 4 mmol) was added to a round-bottomed flask containing 10 mL 6N HCl. The solution was vigorously stirred at 0 °C for 10 min. Sodium nitrite (280 mg, 4 mmol) dissolved in 2 mL water was added dropwise to the aniline solution, and the resulting mixture was stirred at 0 °C for an additional 1 h. To a beaker containing SnCl_2_ (1.9 g, 10 mmol) was added 10 mL of concentrated HCl, and the solution was sonicated and cooled to 0 °C. The SnCl_2_ solution was then added dropwise to the diazonium salt solution, and the mixture was stirred for additional 90 min at room temperature. Then, sodium hydroxide (7.2 g, 0.18 mol) dissolved in 20 mL water was added to quench the reaction. The solution was extracted with 20 mL of diethyl ether three times, and the organic layers were combined and concentrated under reduced pressure to give the crude product, **5** (300 mg, 56%), which was further purified by recrystallization in hot ethanol. The spectral data were in accordance with those reported in the literature [31]. **5**: ^1^H NMR (500 MHz, DMSO-d_6_) δ 7.27–7.20 (m, 2H), 7.08 (d, *J* = 1.1 Hz, 1H), 7.01 (dd, J = 8.4, 1.1 Hz, 1H), 6.93 (d, *J* = 7.4 Hz, 1H), 4.12 (s, 2H). ^13^C NMR (126 MHz, DMSO-*d*_6_) δ 152.9, 129.7, 119.7, 119.4, 116.1, 113.2, 111.4.

9-Bromo-8-methoxy-6,6-dimethyl-6,11-dihydro-5*H*-benzo[*b*]carbazole-3-carbonitrile, **6a**. A mixture of **3** (2.8 g, 9.89 mmol), **5** (1.58 g, 11.9 mmol), and trifluoroacetic acid (48 mL) was refluxed at 80 °C for 2 h. The reaction mixture was cooled and neutralized by saturated aqueous Na_2_CO_3_, then by NaHCO_3_ and extracted with EA. The combined organic layers were washed with brine, dried over Na_2_SO_4,_ and concentrated under reduced pressure. The residue was resuspended in EA and precipitated solid (undesired regioisomer) was filtered off. The filtrate was evaporated under reduced pressure and purified by column chromatography (isocratic elution PS–EA 4:1) to yield **6a** as an orange solid (1.9 g, 50%). **6a**: ^1^H NMR (500 MHz, acetone-*d*_6_) δ 10.79 (s, 1H), 7.75 (s, 1H), 7.67 (d, *J* = 8.1 Hz, 1H), 7.55 (s, 1H), 7.34 (dd, *J* = 8.2, 1.2 Hz, 1H), 7.31 (s, 1H), 4.08 (s, 2H), 3.96 (s, 3H), 1.77 (s, 6H). ^13^C NMR (126 MHz, acetone-d_6_) δ 155.5, 145.4, 144.8, 136.6, 134.2, 131.2, 130.6, 127.7, 122.6, 119.8, 116.2, 111.1, 110.1, 106.8, 104.1, 56.7, 37.1, 31.1, 25.9. ESI-MS, negative mode: *m/z* calcd mass for C_20_H_16_BrN_2_O [M − H]^−^ = 379.04, was found 380.85.

9-Bromo-8-methoxy-6,6-dimethyl-6,11-dihydro-5*H*-benzo[*b*]carbazole-1-carbonitrile, **6b:**
^1^H NMR (500 MHz, DMSO) δ 11.68 (s, 1H), 7.68 (d, J = 7.9 Hz, 1H), 7.54 (s, 1H), 7.46 (d, J = 7.1 Hz, 1H), 7.26 (s, 1H), 7.21 (t, J = 7.6 Hz, 1H), 4.23 (s, 2H), 3.91 (s, 3H), 1.70 (s, 6H). ^13^C NMR (126 MHz, DMSO) δ 154.1, 143.7, 143.6, 136.4, 133.0, 126.3, 125.7, 124.7, 120.8, 119.4, 116.2, 110.4, 108.9, 103.2, 99.8, 56.4, 35.9, 30.8, 25.3.

9-Bromo-8-methoxy-6,6-dimethyl-11-oxo-6,11-dihydro-5*H*-benzo[*b*]carbazole-3-carbonitrile, **7**. To a solution of **6a** (2.1 g, 4.96 mmol) in THF (21.4 mL) and water (2.14 mL) was gradually added DDQ (3.38 g, 14.9 mmol). After stirring for 3 h at 0 °C, the reaction mixture was evaporated under reduced pressure and the residue was dissolved in EA. The solution was washed with aqueous NaOH, water, and brine, dried over Na_2_SO_4_, filtered, and concentrated under reduced pressure. The residue was purified with column chromatography (gradient elution PS–EA 3:1 to 1:2) to afford **7** (1.58 g, 81%) as yellow solid. ^1^H-NMR data were in accordance with those reported in the literature [31]. **7**: ^1^H NMR (500 MHz, acetone-*d*_6_) δ 8.42 (s, 1H), 8.41 (s, 1H), 7.92 (d, *J* = 0.6 Hz, 1H), 7.57 (dd, *J* = 8.1, 1.4 Hz, 1H), 7.50 (s, 1H), 4.09 (s, 3H), 1.91 (s, 6H). ^13^C NMR (126 MHz, acetone-*d*_6_) δ 178.6, 160.2, 159.9, 150.7, 136.9, 131.8, 128.9, 127.3, 125.7, 123.0, 120.4, 117.0, 111.4, 110.8, 110.6, 106.8, 57.2, 38.1, 30.4. ESI-MS, negative mode: *m/z* calcd mass for C_20_H_14_BrN_2_O_2_ [M − H]^−^ = 393.02, was found 394.85.

9-Ethyl-8-methoxy-6,6-dimethyl-11-oxo-6,11-dihydro-5*H*-benzo[*b*]carbazole-3-carbonitrile, **8**. To a degassed mixture of **7** (1.52 g, 3.85 mmol), Cs_2_CO_3_ (3.76 g, 11.5 mmol), Pd(dppf)Cl_2_ (57.7 mg, 0.08 mmol) and DMF (55.7 mL) in a Schlenk flask under argon atmosphere was added a solution of triethylborane in THF (11.5 mL, 1M solution, 11.5 mmol) and the reaction was stirred at 86 °C overnight. The reaction mixture was then cooled, diluted with EA, washed several times with water, dried over Na_2_SO_4_, filtered, and concentrated under reduced pressure. The residue was purified by column chromatography (gradient elution PS–EA 2:1 to 1:2) to yield **8** (870 mg, 66%) as yellow crystalline solid. **8**: ^1^H NMR (500 MHz, acetone-d_6_) δ 11.69 (s, 1H), 8.47 (d, *J* = 8.1 Hz, 1H), 8.12 (s, 1H), 7.90 (s, 1H), 7.57 (dd, *J* = 8.1, 0.5 Hz, 1H), 7.34 (s, 1H), 4.02 (s, 3H), 2.71 (q, *J* = 7.5 Hz, 2H), 1.88 (s, 6H), 1.24 (t, *J* = 7.5 Hz, 3H). ^13^C NMR (126 MHz, acetone-*d*_6_) δ 180.2, 161.9, 160.2, 149.1, 136.8, 132.2, 129.1, 127.4, 125.7, 125.5, 123.0, 120.6, 116.8, 111.1, 108.3, 106.4, 56.2, 37.8, 30.7, 23.6, 14.3. ESI-MS, positive mode: *m/z* calcd mass for C_22_H_20_N_2_O_2_ [M] ^+^ = 344.15, was found 344.95.

9-Ethyl-8-hydroxy-6,6-dimethyl-11-oxo-6,11-dihydro-5*H*-benzo[*b*]carbazole-3-carbonitrile, **9**. Pyridine (141 μL, 1.74 mmol) was slowly added to a hydrochloric acid solution in 1,4-dioxane (436 μL, 4 M, 1.74 mmol) at 0 °C under argon. The mixture was stirred for 5 min and then concentrated under reduced pressure to form white highly hygroscopic pyridinium chloride salt. Both **8** (12 mg, 34.8 μmol) and pyridinium chloride (1.74 mmol) were quickly transferred to a sealed tube and stirred at 170 °C overnight. The reaction mixture was dissolved in water, neutralized by saturated aqueous NaHCO_3_, and extracted with EA three times. The combined organic layers were washed with aqueous NaHCO_3_ and brine, dried over Na_2_SO_4_, filtered, and concentrated under reduced pressure. The residue was purified by flash column chromatography (gradient elution of PS–EA 2:1 to 1:2) to yield **9** as a beige solid (9.5 mg, 83%). **9**: ^1^H NMR (500 MHz, DMSO-d_6_) δ 12.67 (s, 1H), 10.20 (s, 1H), 8.32 (dd, *J* = 8.1, 0.4 Hz, 1H), 7.98 (d, *J* = 0.6 Hz, 1H), 7.94 (s, 1H), 7.59 (dd, *J* = 8.1, 1.4 Hz, 1H), 7.10 (s, 1H), 2.62 (q, *J* = 7.5 Hz, 2H), 1.70 (s, 6H), 1.19 (t, *J* = 7.5 Hz, 3H). ^13^C NMR (126 MHz, DMSO-d_6_) δ 179.2, 159.4, 159.3, 147.9, 135.6, 129.4, 127.8, 126.8, 124.7, 123.3, 121.7, 120.2, 116.4, 111.8, 109.4, 104.4, 36.2, 30.3, 22.5, 13.9. ESI-MS, positive mode: *m/z* calcd mass for C_21_H_18_N_2_O_2_ [M + H] ^+^ = 330.14, was found 330.90.

3-Cyano-9-ethyl-6,6-dimethyl-11-oxo-6,11-dihydro-5*H*-benzo[*b*]carbazol-8-yl trifluoromethanesulfonate, **10**. A suspension of **9** (480 mg, 1.45 mmol) and K_2_CO_3_ (402 mg, 2.91 mmol) in DMF (13.7 mL) was slowly (portionwise) treated with *N*-phenyl-bis(trifluoromethanesulfonimide) (1.04 g, 2.91 mmol) at 0 °C. After 24 h, the reaction mixture was diluted with EA and poured into water. The layers were partitioned, and the organic phase washed with brine, dried over Na_2_SO_4_, and concentrated in vacuo. The residue was purified by silica gel chromatography (gradient elution PS–EA 3:1 to 1:1) to afford **10** (540 mg, 80% yield) as a yellow solid. **10**: ^1^H NMR (500 MHz, acetone-*d*_6_) δ 11.87 (s, 1H), 8.44 (d, *J* = 8.1 Hz, 1H), 8.37 (s, 1H), 7.94 (d, *J* = 1.3 Hz, 1H), 7.84 (s, 1H), 7.60 (dd, *J* = 8.1, 1.3 Hz, 1H), 2.89 (q, *J* = 7.6 Hz, 2H), 1.91 (s, 6H), 1.36 (t, *J* = 7.6 Hz, 3H). ^13^C NMR (126 MHz, acetone-*d*_6_) δ 178.8, 160.3, 151.5, 149.1, 137.1, 136.3, 132.9, 129.5, 128.9, 126.1, 123.2, 120.7, 120.4, 119.7, 117.2, 111.2, 107.2, 37.9, 30.4, 23.4, 14.2. ESI-MS, positive mode: *m/z* calcd mass for C_22_H_17_F_3_N_2_O_4_S [M] ^+^ = 462.08, was found 462.95.

1-(Tetrahydro-2*H*-pyran-2-yl)-1*H*-pyrazole, **12**. A mixture of pyrazole (1.43 g, 0.21 mol), 3,4-dihydro-2*H*-pyran (29 mL, 0.32 mol), and trifluoroacetic acid (80 μL, 1 mmol) was refluxed for 2 h. After quenching with sodium hydride (30 μg, 1.26 mmol), the mixture was subjected to column chromatography (gradient elution PS–EA 20:1 to 5:1) to yield **12** (3.2 g, 95%) as a colorless oil. The spectral data were in accordance with those reported in the literature [32]. **12**: ^1^H NMR (500 MHz, CDCl_3_) δ 7.61 (d, *J* = 2.4 Hz, 1H), 7.56 (d, *J* = 1.4 Hz, 1H), 6.32–6.29 (m, 1H), 5.40 (dd, *J* = 9.7, 2.5 Hz, 1H), 4.06 (dd, *J* = 12.2, 2.6 Hz, 1H), 3.73–3.67 (m, 1H), 2.23–2.05 (m, 3H), 1.77–1.55 (m, 3H).

(1-(Tetrahydro-2*H*-pyran-2-yl)-1*H*-pyrazol-3-yl)boronic acid, **S1**. To a solution of 1-(tetrahydropyran-2-yl)-1*H*-pyrazole (7.64 g, 0.052 mol) in THF (50 mL), cooled to −78 °C, was added n-butyllithium (1.6 M in hexane, 32 mL, 0.052 mol) dropwise. Bulky precipitate formed and the mixture turned yellow. After 30 min, trimethyl borate (5.88 mL, 0.058 mol) was added over 10 min, and the reaction mixture was stirred at −78 °C for 1 h. The reaction was quenched with hydrochloric acid (2 M in water, 0.104 mol, 52 mL) and was left to return to ambient temperature in 1 h. The mixture was extracted with EA and the organic layers were dried over Na_2_SO_4_ and concentrated in vacuo. The residue was crystallized with PS–EA (12:1) to yield **S1** (3.94 g, 40%) as a white solid. **S1** was used to the next step without further purification.

*3-Borono-1H-pyrazol-1-ium trifluoroacetate,***13**. To a solution of **S1** (110 mg, 0.56 mmol) in dichloromethane (1 mL), trifluoroacetic acid (0.5 mL) was added at room temperature, and the reaction solution was stirred for 24 h. The reaction solution was concentrated, and the residue was crystallized with PS–DCM–dioxane (5:10:1) to afford **13** (70 mg 55%) as white powder. The spectral data were in accordance with those reported in the literature [33]. **13**: 1H NMR (500 MHz, DMSO-d_6_) δ 8.28 (s, 2H), 7.52 (d, *J* = 1.6 Hz, 1H), 6.70 (d, *J* = 1.7 Hz, 1H).

9-Ethyl-6,6-dimethyl-11-oxo-8-(1*H*-pyrazol-3-yl)-6,11-dihydro-5*H*-benzo[*b*]carbazole-3-carbonitrile, **JH-VII-139-1**. A degassed mixture of **10** (76 mg, 0.164 mmol), K_2_CO_3_ (182 mg, 1.31 mmol), Pd(dppf)Cl_2_ (6 mg, 0.008 mmol), **13** (85 mg, 0.378 mmol), tBuXPhos (5.6 mg, 0.013 mmol), dioxane (5.06 mL) and H_2_O (1.26 mL) was stirred in a microwave tube at 70 °C in microwave for two hours. The reaction mixture was then cooled, diluted with EA, washed three times with water, dried over Na_2_SO_4_, filtered, and concentrated under reduced pressure. The residue was purified by flash column chromatography (gradient elution PS–EA 1:1, 1:2, 1:4) to yield **JH-VII-139-1** (50 mg, 80%) as brown crystalline solid. **JH-VII-139-1**: ^1^H NMR (500 MHz, DMSO-*d*_6_) δ 13.08 (s, 1H), 8.35 (d, *J* = 8.1 Hz, 1H), 8.13 (s, 1H), 8.03 (s, 1H), 7.90 (s, 1H), 7.83 (s, 1H), 7.63 (d, *J* = 8.1 Hz, 1H), 6.65 (s, 1H), 2.94 (m, 2H), 1.80 (s, 6H), 1.19 (t, *J* = 7.5 Hz, 3H). ^13^C NMR (126 MHz, DMSO-*d*_6_) δ 180.3, 160.6, 149.2 146.2, 141.7, 137.5, 136.9, 132.4, 131.8, 129.1, 128.4, 127.6, 125.7, 123.1, 120.5, 117.0, 111.4, 106.7, 106.1, 37.4, 30.3, 27.2, 15.8. ESI-MS, negative mode: *m/z* calcd mass for C_24_H_19_N_4_O [M − H]^−^ = 379.15, was found 379.00.

(Ethane-1,2-diylbis(oxy))bis(ethane-2,1-diyl) bis(4-nitrophenyl) dicarbonate, **16**. To a stirred solution of triethyleneglycol (62 mg, 0.412 μmol) in DCM (1.3 mL) and triethylamine (344 μL, 2.47 mmol), at 0 °C under argon, was added 4-nitrophenyl chloroformate (250 mg, 1.23 mmol). As the solution turned yellow, the mixture was stirred at room temperature for 2 h. The solvent was removed under reduced pressure, and the residue was chromatographed on silica gel with PS–EA (4:1, 3:1, 2:1, 1:1, EA) to afford **15** (198 mg, 76%) as a colorless oil. The spectral data were in accordance with those reported in the literature [34]. **16**: 1H NMR (500 MHz, CDCl_3_) δ 8.27 (d, *J* = 9.0 Hz, 4H), 7.38 (d, *J* = 9.0 Hz, 4H), 4.45 (dd, *J* = 5.4, 3.8 Hz, 4H), 3.83 (dd, *J* = 5.3, 4.0 Hz, 4H), 3.74 (s, 4H).

2-(2-(2-(((4-Nitrophenoxy)carbonyl)oxy)ethoxy)ethoxy)ethyl 3-(3-cyano-9-ethyl-6,6-dimethyl-11-oxo-6,11-dihydro-5*H*-benzo[*b*]carbazol-8-yl)-1*H*-pyrazole-1-carboxylate, **17**. To a stirred solution of (ethane-1,2-diylbis(oxy))bis(ethane-2,1-diyl) bis(4-nitrophenyl) dicarbonate **16** (53 mg, 0.11 mmol) and triethylamine (31 μL, 0.22 mmol) in dry DMF (1.2 mL) was added dropwise a solution of **JH-VII-139-1** (12 mg, 31.5 μmol) in DMF (1.2 mL) at 0 °C under argon. The reaction mixture was stirred at room temperature for 24 h and then, the solvent was evaporated. The residue was chromatographed on silica gel with PS–EA (gradient elution 1:1, 1:2, 1:3, 1:4) to give **17** (22 mg, 55% yield) as a yellow solid. **17**: ^1^H NMR (500 MHz, CDCl_3_) δ 9.52 (s, 1H), 8.54 (d, J = 8.1 Hz, 1H), 8.35 (s, 1H), 8.29–8.23 (m, 3H), 7.77 (s, 1H), 7.72 (s, 1H), 7.58 (dd, J = 8.2, 1.2 Hz, 1H), 7.40–7.35 (m, 2H), 6.65 (d, J = 2.8 Hz, 1H), 4.68–4.65 (m, 2H), 4.44–4.41 (m, 2H), 3.93–3.90 (m, 2H), 3.83–3.80 (m, 2H), 3.76–3.70 (m, 4H), 2.87 (q, J = 7.6 Hz, 2H), 1.75 (s, 6H), 1.24 (t, J = 7.6 Hz, 3H). ^13^C NMR (126 MHz, CDCl_3_) δ 180.4, 159.0, 156.2, 155.6, 152.6, 149.5, 145.6, 144.7, 142.0, 135.8, 135.5, 132.1, 131.9, 128.5, 127.7, 127.4, 125.9, 125.5, 123.1, 121.9, 120.2, 115.8, 111.5, 110.5, 106.5, 70.9, 68.9, 68.9, 68.4, 67.5, 36.5, 30.7, 26.5, 15.5. ESI-MS, negative mode: *m/z* calcd mass for C_38_H_38_N_7_O_8_ [M] ^+^ = 721.24, was found 719.95.

Conjugate **geo75**. To a stirred solution of **17** (3.48 mg, 4.83 μmol) and c(RGDyK) (2.3 mg, 3.71 μmol) in dry DMF (1.15 mL), under argon, was added DIPEA (2.59 μL, 14.8 μmol), and the mixture was stirred at room temperature for 24 h. The solvent was evaporated, and the product was purified by HPLC (method 1, Appendix A) to yield **geo75** as white solid (2.5 mg, 56%). **geo75**: ^1^H NMR (500 MHz, DMSO-*d*_6_) δ 13.03 (s, 1H), 9.09 (s, 1H), 8.47 (d, *J* = 2.7 Hz, 1H), 8.33 (d, *J* = 8.1 Hz, 2H), 8.27 (s, 1H), 8.20 (d, *J* = 9.3 Hz, 2H), 8.16 (s, 2H), 8.12 (d, *J* = 9.2 Hz, 1H), 8.01 (s, 1H), 7.91 (s, 1H), 7.66–7.59 (m, 1H), 7.13 (s, 3H), 7.01 (d, *J* = 2.7 Hz, 1H), 6.92 (d, *J* = 8.4 Hz, 2H), 6.58 (d, *J* = 8.4 Hz, 2H), 4.54 (m, 2H), 4.47 (s, 1H), 4.23 (m, 1H), 4.13 (m, 1H), 3.98 (s, 2H), 3.80–3.76 (m, 2H), 3.62–3.57 (m, 2H), 3.53 (s, 4H), 3.16 (m, 3H), 3.04 (dd, *J* = 12.3, 6.2 Hz, 2H), 2.95–2.85 (m, 4H), 2.70 (m, 1H), 2.50 (m, 2H)2.11 (m, 1H), 1.78 (s, 6H), 1.62 (m, 3H), 1.43 (s, 4H), 1.30 (d, *J* = 7.1 Hz, 2H), 1.16 (t, *J* = 7.5 Hz, 3H), 1.10 (m, 1H). ^13^C NMR (126 MHz, DMSO-d_6_) δ 178.9, 173.5, 172.0, 170.9, 170.8, 170.6, 167.8, 160.4, 156.8, 156.1, 155.5, 155.0, 148.8, 145.6, 140.9, 135.9, 135.0, 132.4, 131.3, 129.8, 128.0, 127.9, 127.7, 127.4, 126.3, 125.0, 121.6, 120.0, 116.6, 114.8, 110.5, 109.5, 104.8, 69.9, 69.7, 69.6, 68.9, 68.0, 67.4, 63.0, 54.9, 53.3, 50.8, 48.6, 40.6, 38.9, 36.4, 31.8, 29.9, 29.0, 28.8, 27.3, 26.0, 24.5, 22.9, 15.3. ESI-MS, positive mode: *m/z* calcd mass for C_59_H_71_N_13_O_15_ [M + H] ^+^ = 1202.52, was found 1202.30.

Bis(2,5-dioxopyrrolidin-1-yl) glutarate, **19** was synthesized and characterized by our group in a previous work [26].

2,5-Dioxopyrrolidin-1-yl 5-(3-(3-cyano-9-ethyl-6,6-dimethyl-11-oxo-6,11-dihydro-5*H*-benzo[b]carbazol-8-yl)-1*H*-pyrazol-1-yl)-5-oxopentanoate, **20**. To a stirred solution of bis(2,5-dioxopyrrolidin-1-yl) glutarate **19** (18.9 mg, 57.8 μmol) and **JH-VII-139-1** (11 mg, 28.9 μmol) in DMF (0.5 mL) was added triethylamine (16.6 μL, 116 μL) under argon. The reaction was then stirred at room temperature for 48 h. The solvent was removed under vacuum and the residue was chromatographed on silica gel with DCM–Acetone (gradient elution 4:0.1 to 4:0.3) to give **20** (8 mg, 47%) as white solid. **20**: ^1^H NMR (500 MHz, DMSO-d_6_) δ 12.84 (s, 1H), 8.57 (d, J = 2.8 Hz, 1H), 8.34 (d, J = 8.1 Hz, 1H), 8.17 (s, 1H), 8.02 (s, 2H), 7.62 (d, J = 8.2 Hz, 1H), 7.14 (d, J = 2.8 Hz, 1H), 3.32–3.36 (m, 2H), 3.00 (q, J = 7.4 Hz, 2H), 2.87 (t, J = 7.4 Hz, 2H), 2.82 (s, 4H), 2.11–2.04 (m, 2H), 1.81 (s, 6H), 1.23 (t, J = 7.5 Hz, 3H). ^13^C NMR (126 MHz, DMSO-d_6_) δ 178.7, 171.1, 170.2, 168.7, 154.6, 145.7, 140.9, 134.5, 131.4, 129.4, 128.0, 126.6, 124.8, 121.6, 120.1, 119.5, 118.1, 117.9, 116.7, 110.9, 109.5, 104.5, 36.5, 32.2, 29.9, 29.4, 26.4, 25.4, 19.1, 15.4. ESI-MS, positive mode: *m/z* calcd mass for C_33_H_30_N_5_O_6_ [M + H] ^+^ = 592.21, was found 592.00.

Conjugate **geo77**. To a stirred solution of **20** (3.82 mg, 6.46 μmol) and c(RGDyK) (4 mg, 6.46 μmol) in DMF (2 mL), under argon, was added DIPEA (4.5 μL, 25.8 μmol) and the mixture was stirred at room temperature for 24 h. The solvent was evaporated and **geo77** was purified by HPLC (method 1, Appendix A) to yield **geo77** as a white solid (3.2 mg, 45%). **geo77**: ^1^H NMR (500 MHz, DMSO-*d*_6_) δ 12.97 (s, 1H), 9.11 (s, 1H), 8.54 (d, *J* = 2.8 Hz, 1H), 8.40 (d, *J* = 8.7 Hz, 1H), 8.35 (d, *J* = 8.1 Hz, 1H), 8.20–8.30 (m, 3H), 8.21–8.12 (m, 3H), 8.03 (s, 1H), 8.01 (s, 1H), 7.91 (t, *J* = 5.6 Hz, 1H), 7.68–7.59 (m, 2H), 7.00–7.25 (m, 3H), 6.93 (d, *J* = 8.5 Hz, 2H), 6.59 (d, *J* = 8.4 Hz, 2H), 4.56–4.46 (m, 2H), 4.33–4.26 (m, 1H), 4.23–4.11 (m, 2H), 3.20–3.16 (m, 4H), 3.09–3.03 (m, 2H), 2.95–3.42 (m, 4H), 2.75–2.69 (m, 1H), 2.21 (t, *J* = 7.4 Hz, 2H), 2.06–2.13 (m, 1H), 1.98–1.90 (m, 2H), 1.81 (d, *J* = 1.7 Hz, 6H), 1.54–1.70 (m, 3H), 1.53–1.42 (m, 2H), 1.41–1.33 (m, 2H), 1.22 (t, *J* = 7.5 Hz, 3H), 1.20–1.11 (m, 2H). ^13^C NMR (126 MHz, DMSO-*d*_6_) δ 178.9, 173.5, 172.0, 171.5, 171.4, 170.9, 170.8, 170.6, 167.7, 160.4, 156.7, 155.5, 154.4, 145.6, 140.9, 135.8, 134.7, 131.2, 129.8, 129.4, 128.3 127.9, 127.6, 126.7, 125.0, 123.9, 121.7, 120.0, 116.5, 114.8, 110.8, 109.5, 104.8, 54.9, 53.3, 50.8, 48.6, 42.9, 40.6, 38.3, 36.4, 35.9, 34.3, 32.9, 31.8, 29.8, 29.0, 28.5, 27.2, 26.5, 24.5, 23.0, 20.2, 15.5. ESI-MS, positive mode: *m/z* calcd mass for C_56_H_66_N_13_O_11_ [M + H] ^+^ = 1096.49, was found 1096.00.

Bis(4-nitrophenyl) ((ethane-1,2-diylbis(oxy))bis(ethane-2,1-diyl))dicarbamate, **22**. 2,2′-(Ethylenedioxy)bis(ethylamine) (1.00 g, 6.75 mmol) and pyridine (2.18 mL, 27.0 mmol) were dissolved in CH_2_Cl_2_ (250 mL). The reaction flask was cooled to 0 °C and then *p*-nitrophenyl chloroformate (5.44 g, 27.0 mmol) was added. The solution was warmed to room temperature while stirring for 24 h. The reaction was filtered, and the solvent was evaporated leaving a white solid. The crude product was dissolved in CH_2_Cl_2_ (150 mL) and washed with aqueous 1 M NaHSO_4_ (3 × 30 mL). The organic layer was dried with Na_2_SO_4,_ and the solvent was evaporated. The resulting solid was crystallized twice from CH_2_Cl_2_–hexanes (1:1) providing **22** (2.19 g, 68%) as a white solid. **22**: ^1^H NMR (500 MHz, acetone-*d*_6_, DMSO-*d*_6_) δ 8.27 (d, *J* = 9.1 Hz, 4H), 7.91 (t, *J* = 5.3 Hz, 2H), 7.43 (d, *J* = 9.2 Hz, 4H), 3.63 (s, 4H), 3.60 (t, *J* = 5.8 Hz, 4H), 3.34 (q, *J* = 5.8 Hz, 4H). ^13^C NMR (126 MHz, acetone-*d*_6_, DMSO-*d*_6_) δ 157.12, 153.93, 144.86, 125.54, 122.86, 70.48, 69.66, 41.28.

4-Nitrophenyl (2-(2-(2-(3-(3-cyano-9-ethyl-6,6-dimethyl-11-oxo-6,11-dihydro-5*H*-benzo[*b*]carbazol-8-yl)-1*H*-pyrazole-1-carboxamido)ethoxy)ethoxy)ethyl)carbamate, **23**. To a stirred solution of bis(4-nitrophenyl) ((ethane-1,2-diylbis(oxy))bis(ethane-2,1-diyl))dicarbamate **22** (30.2 mg, 63.1 μmol) and triethylamine (17.6 μL, 126 μmol) in dry DMF (0.8 mL) was added dropwise a solution of JH-VII-139-1 (1eq) in dry DMF (0.8 μL) at 0 °C under Ar. The reaction was stirred at 25 °C overnight. Subsequently, the solvent was evaporated, and the residue was chromatographed on silica gel with DCM–Acetone (gradient elution 4:0.1, 4:0.2, 4:0.3, 4:0.4, 4:0.6) to give **23** (15 mg, 66%) as a beige solid. **23**: ^1^H NMR (500 MHz, acetone-d_6_) δ 11.79 (s, 1H), 8.49 (d, J = 8.1 Hz, 1H), 8.39 (d, J = 2.6 Hz, 1H), 8.29 (s, 1H), 8.22 (d, J = 9.1 Hz, 2H), 8.01 (s, 1H), 7.93 (s, 1H), 7.90 (s, 1H), 7.61 (d, J = 8.1 Hz, 1H), 7.37 (d, J = 9.1 Hz, 2H), 6.99 (s, 1H), 6.88 (d, J = 2.6 Hz, 1H), 3.74 (t, J = 5.5 Hz, 2H), 3.69–3.60 (m, 8H), 3.35 (q, J = 5.6 Hz, 2H), 3.02 (q, J = 7.5 Hz, 2H), 1.91 (s, 6H), 1.26 (d, J = 7.5 Hz, 3H). ^13^C NMR (126 MHz, acetone-d_6_) δ 178.9, 159.4, 156.3, 153.2, 152.9, 149.2, 145.2, 141.0, 135.8, 135.4, 131.5, 128.9, 127.9, 127.7, 127.6, 126.5, 124.6, 124.5, 121.9, 121.7, 119.3, 115.8, 110.2, 108.7, 105.6, 69.9, 69.0, 40.6, 39.7, 36.3, 29.4, 26.1, 14.6. ESI-MS, positive mode: *m/z* calcd mass for C_38_H_38_N_7_O_8_ [M + H] ^+^ = 720.27, was found 720.00.

Conjugate **geo85**. To a stirred solution of **23** (3 mg, 4.2 μmol) and c(RGDyK) (2 mg, 3.23 μmol) in dry DMF (1 mL), under argon, was added DIPEA (2.25 uL, 12.9 μmol) and the mixture was stirred at room temperature for 24 h. The solvent was evaporated and **geo85** was isolated by HPLC (method 1, Appendix A) to yield **geo85** (3.4 mg, 88%) as a white solid. **geo85**: ^1^H NMR (500 MHz, DMSO-*d*_6_) δ 13.05 (s, 1H), 9.12 (s, 1H), 8.44 (d, *J* = 2.7 Hz, 1H), 8.40–8.30 (m, 3H), 8.27 (s, 1H), 8.23 (d, *J* = 9.2 Hz, 2H), 8.17 (s, 1H), 8.13–8.08 (m, 3H), 8.02 (s, 1H), 7.95 (s, 1H), 7.65 (d, *J* = 9.9 Hz, 1H), 7.63 (dd, *J* = 8.2, 1.3 Hz, 1H), 7.30–6.98 (br, 3H) 6.94 (d, *J* = 8.5 Hz, 2H), 6.91 (d, *J* = 2.7 Hz, 1H), 6.60 (d, *J* = 8.4 Hz, 2H), 6.00 (t, *J* = 5.5 Hz, 1H), 5.86 (t, *J* = 5.7 Hz, 1H), 4.60–4.53 (m, 1H), 4.52–4.47 (m, 1H), 4.32–4.25 (m, 1H), 4.23–4.19 (m, 1H), 4.19–4.13 (m, 1H), 3.62–3.54 (m, 4H), 3.52–3.46 (m, 4H), 3.20 (dd, *J* = 14.2, 4.8 Hz, 2H), 3.12–3.05 (m, 4H), 2.97–2.92 (m, 2H), 2.91–2.86 (m, 2H), 2.74 (dd, *J* = 16.9, 2.8 Hz, 1H), 2.53–2.51 (m, 2H), 2.10 (d, *J* = 14.2 Hz, 1H), 1.80 (d, *J* = 2.5 Hz, 6H), 1.73–1.66 (m, 2H), 1.65–162 (m, 1H), 1.57–1.49 (m, 2H), 1.48–1.41 (m, 2H), 1.35–1.30 (m, 2H), 1.17 (t, *J* = 7.5 Hz, 3H), 1.15–1.10 (m, 1H). ^13^C NMR (126 MHz, DMSO-*d*_6_) δ 179.0, 173.7, 172.0, 171.0, 170.7, 170.5, 167.6, 160.4, 158.1, 156.7, 155.5, 153.0, 149.4, 145.5, 140.8, 135.9, 135.5, 131.1, 129.8, 129.6, 128.3, 128.0, 127.6, 126.0, 124.9, 121.6, 120.0, 116.6, 114.8, 109.5, 109.3, 104.8, 70.1, 69.5, 55.0, 53.2, 50.7, 48.6, 42.9, 40.6, 36.4, 35.8, 31.9, 29.9, 29.8, 29.4, 27.3, 26.0, 24.6, 23.0, 15.19; 4 peaks are missing due to overlapping. ESI-MS, positive mode: *m/z* calcd mass for C_59_H_74_N_15_O_13_ [M + H] ^+^ = 1200.55, was found 1199.80.

(*S*)-14-(4-((tert-butoxycarbonyl)amino)butyl)-1-(3-(3-cyano-9-ethyl-6,6-dimethyl-11-oxo-6,11-dihydro-5*H*-benzo[*b*]carbazol-8-yl)-1*H*-pyrazol-1-yl)-1,12-dioxo-5,8-dioxa-2,11,13-triazapentadecan-15-oic acid, **24**. To a stirred solution of **23** (11 mg, 15.3 μmol) and Nε-Boc-L-lysine (3.8 mg, 15.3 μmol) in DMF (0.8 mL) was added triethylamine (8.5 μL, 61 μmol) at ambient temperature, under Ar. The mixture was stirred at room temperature for 24 h. Subsequently, the solvent was evaporated, and the product was purified with flash column chromatography (gradient elution DCM–MeOH 10:1, 10:2, 10:3) to yield **24** (8 mg, 63%) as white solid. **24**: ^1^H NMR (500 MHz, DMSO-d_6_) δ 8.40 (d, *J* = 2.7 Hz, 1H), 8.33 (d, *J* = 8.1 Hz, 1H), 8.23 (t, *J* = 4.9 Hz, 1H), 8.17 (s, 1H), 8.10 (s, 1H), 7.94 (s, 1H), 7.58 (dd, *J* = 8.1, 1.3 Hz, 1H), 6.87 (d, *J* = 2.7 Hz, 1H), 6.57 (s, 1H), 6.07–6.01 (m, 2H), 3.95 (dd, *J* = 12.6, 7.3 Hz, 1H), 3.61 (t, *J* = 5.7 Hz, 2H), 3.58 (dd, *J* = 5.7, 4.0 Hz, 2H), 3.53 (dd, *J* = 6.1, 4.1 Hz, 2H), 3.51–3.46 (m, 3H), 3.38 (t, *J* = 5.7 Hz, 2H), 3.13–3.07 (m, 4H), 2.96–2.91 (m, 2H), 2.87 (dd, *J* = 12.9, 6.8 Hz, 2H), 1.80 (d, *J* = 3.7 Hz, 6H), 1.66–1.59 (m, 2H), 1.53–1.46 (m, 2H), 1.37–1.31 (m, 11H), 1.18 (t, *J* = 7.6 Hz, 3H). ^13^C NMR (126 MHz, DMSO-d_6_) δ 178.7, 174.8, 160.4, 157.6, 155.3, 152.7, 149.2, 145.4, 140.4, 136.1, 135.2, 131.1, 129.4, 127.7, 127.6, 125.9, 124.5, 121.4, 119.8, 116.6, 109.3, 109.0, 104.4, 77.1, 70.0, 69.4, 68.5, 53.1, 36.3, 32.2, 31.0, 29.8, 29.7, 28.7, 28.1, 25.7, 22.5, 21.8, 14.9. ESI-MS, positive mode: *m/z* calcd mass for C_43_H_55_N_8_O_9_ [M + H] ^+^ = 827.40, was found 826.95.

Compound **25**. In a stirred solution of **24** (7 mg, 8.32 μmol) and NHS (1.44 mg, 12.5 μmol) in dry DMF (0.5 mL), was added EDCi (4.8 mg, 25 μmol) under argon at room temperature. After 3 h and upon starting material consumption (TLC monitoring), c(RGDyK) (3.6 mg, 5.83 μmol) was added, and the reaction was stirred for 48 h at room temperature. The mixture was then concentrated under reduced pressure and the residue purified by HPLC (method 1, Appendix A) to afford conjugate **25** (3.5 mg, 30%) as white solid. **25**: ^1^H NMR (500 MHz, DMSO) δ 13.10 (s, 1H), 9.15 (s, 1H), 8.49–8.31 (m, 6H), 8.24–8.16 (m, 6H), 8.04 (s, 1H), 7.96 (s, 1H), 7.91 (s, 2H), 7.70–7.62 (m, 2H), 7.27 (s, 2H), 6.98–6.85 (m, 3H), 6.77–6.70 (m, 1H), 6.60 (d, *J* = 8.2 Hz, 2H), 6.16 (d, *J* = 8.4 Hz, 2H), 6.10 (s, 1H), 4.59–4.46 (m, 2H), 4.32–4.26 (m, 1H), 4.22–4.10 (m, 2H), 4.07–4.00 (m, 1H), 3.62–3.54 (m, 4H), 3.53–3.45 (m, 4H), 3.20 (d, *J* = 9.3 Hz, 3H), 3.14–3.04 (m, 6H), 3.00–2.91 (m, 4H), 2.89–2.83 (m, 2H), 2.73 (d, *J* = 15.5 Hz, 1H), 2.57–2.53 (m, 2H), 2.12–2.06 (m, 1H), 1.80 (s, 6 H), 1.72–1.58 (m, 3H), 1.57–1.44 (m, 6H), 1.40–1.29 (m, 13H), 1.23–1.10 (m, 6H). ^13^C NMR (126 MHz, DMSO) δ 179.1, 173.7, 172.5, 172.0, 171.1, 170.1, 170.6, 167.7, 160.5, 157.6, 156.8, 155.5, 153.0, 149.5, 145.5, 140.8, 135.9, 135.5, 131.2, 129.9, 129.7, 128.4, 128.1, 127.7, 126.1, 125.0, 121.7, 120.1, 118.2, 116.7, 114.8, 109.5, 109.4, 104.8, 77.3, 70.1, 69.6, 69.6, 68.7, 55.0, 52.9, 50.8, 48.6, 40.4, 38.2, 36.4, 33.2, 29.9, 29.9, 29.3, 28.5, 28.3, 26.0, 24.6, 23.0, 22.6, 15.2; 3 peaks are missing due to overlapping. ESI-MS, negative mode: *m/z* calcd mass for C_70_H_93_N_17_O_16_ [M − H]^−^ = 1426.70, was found 1426.10.

Conjugate **geo107**. Compound **25** (1.8 mg, 1.26 μmol) was dissolved in a H_2_O/ACN/TESH/TFA (0.36 mL, 80/180/20/80) mixture and stirred for 2 h at room temperature. The mixture was then freeze dried and subjected to HPLC (method 1, Appendix A) to yield **geo107** (1.5mg, 90%) as beige solid. **geo107**: ^1^H NMR (500 MHz, DMSO-*d*_6_) δ ^1^H NMR (500 MHz, DMSO-*d*_6_) δ 9.02 (s, 1H), 8.46–8.36 (m, 4H), 8.34 (d, *J* = 8.1 Hz, 1H), 8.28 (d, *J* = 8.4 Hz, 1H), 8.20 (d, *J =* 9.7 Hz, 1H), 8.17 (s, 1H), 8.10 (d, *J* = 9.2 Hz, 1H), 8.07–7.89 (m, 5H), 7.66 (d, *J* = 9.1 Hz, 1H), 7.63 (d, *J* = 8.7 Hz, 1H), 7.49–7.24 (br, 3H), 6.96–6.90 (m, 3H), 6.86 (d, *J* = 8.4 Hz, 1H), 6.60 (d, *J* = 8.3 Hz, 2H), 6.26–6.14 (m, 2H), 4.58–4.46 (m, 2H), 4.35–4.28 (m, 1H), 4.20–4.11 (m, 2H) 4.08–0.2 (m, 1H), 3.60–3.54 (m, 4H), 3.53–3.44 (m, 4H), 3.26–3.21 (m, 3H), 3.14–3.03 (m, 6H), 2.98–2.80 (m, 4H), 2.74–2.68 (m, 3H), 2.10–2.04 (m, 1H), 1.80 (s, 6H), 1.62–1.55 (m, 3H), 1.54–1.42 (m, 6H), 1.27–1.20 (m, 4H), 1.19–1.13 (m, 6H). ^13^C NMR (126 MHz, DMSO) δ 179.0, 173.9, 172.3, 170.8, 166.6, 166.4, 166.3, 160.6, 157.6, 157.1, 157.0, 155.8, 155.5, 153.0, 149.5, 145.6, 140.8, 136.1, 135.5, 131.2, 129.9, 129.8, 129.7, 128.1, 127.7, 126.1, 125.0, 121.6, 120.1, 116.7, 115.1, 114.9, 109.5, 109.3, 104.8, 70.1, 69.6, 68.7, 55.8, 52.7, 49.8, 46.0, 40.5, 38.9, 36.5, 32.9, 29.9, 28.5, 27.6, 26.0, 25.2, 25.1, 22.2, 22.1, 15.2; 3 peaks are missing due to overlapping. ESI-MS, positive mode: *m/z* calcd mass for C_65_H_86_N_17_O_14_ [M + H] ^+^ = 1328.65, was found 1327.9.

3′,6′-Dihydroxy-3-oxo-3*H*-spiro[isobenzofuran-1,9′-xanthene]-5(6)-carboxylic acid, **28**. 1,3-Dioxo-1,3-dihydroisobenzofuran-5-carboxylic acid **26** (1 g, 5.1 mmol) and resorcinol **27** (1.15 g, 10.4 mmol) in methanesulfonic acid (5 mL) were placed in a pressure tube equipped with a magnetic stirrer bar. The reaction mixture was stirred at 80 °C overnight and then poured to cold water (500 mL) with stirring and filtered. The solid residue was heated at reflux in EtOH (200 mL), H_2_O was added until precipitation, and the mixture cooled to room temperature, filtered, and freeze-dried for 24 h to yield 1.9 g (97%) of crude 5(6)-carboxyfluorescein as an orange powder. The crude compound was used without further purification. The spectral data were in accordance with those reported in the literature [35]. **28**: ^1^H NMR (500 MHz, DMSO-*d*_6_) δ 10.14 (s, 2H), 8.40 (s, 1H), 8.29 (dd, *J* = 8.0, 1.4 Hz, 1H), 8.22 (d, *J* = 8.0 Hz, 0.7H), 8.11 (d, *J* = 8.0 Hz, 0.7H), 7.65 (s, 0.7H), 7.39 (d, *J* = 8.0 Hz, 1H), 6.70–6.65 (m, 2.2H), 6.66–6.58 (m, 3.2H), 6.56–6.52 (m, 4.8H), regioisomer ratio present: 60–40 (5/6).

2,5-Dioxopyrrolidin-1-yl 3′,6′-dihydroxy-3-oxo-3*H*-spiro[isobenzofuran-1,9′-xanthene]-5(6)-carboxylate, **29**. 5(6)-Carboxyfluorescein **28** (200 mg, 0.531 mmol) was dissolved in dry DMF (2.5 mL). EDCi (306 mg, 1.59 mmol) and *N*-hydroxysuccinimide (122 mg, 1.06 mmol) were added to the mixture. The reaction was stirred under argon for 2 h and after completion, was washed with aqueous KH_2_PO_4_ three times (EA was used in moderation as additional organic solvent, since EA does not dissolve fluorescein derivatives). Organic layers were dried with Na_2_SO_4_ and concentrated under reduced pressure. The residue was then subjected to column chromatography (gradient elution EA, EA–MeOH, 10–1, 10–2) and yielded **29** as dark brown solid (120 mg, 48%). The spectral data were in accordance with those reported in the literature [35]. **29**: ^1^H NMR (500 MHz, DMSO-*d*_6_) δ 10.18 (s, 2H), 8.54 (s, 1H), 8.43 (dd, *J* = 8.1, 1.6 Hz, 1H), 7.56 (d, *J* = 8.1 Hz, 1H), 6.72–6.65 (m, 4H), 6.57–6.53 (m, 2H), 2.93 (s, 4H), regioisomer ratio present: 72–28 (5/6). ESI-MS, positive mode: *m/z* calcd mass for C_25_H_15_NO_9_ [M] ^+^ = 473.07 was found 473.85.

5(6)-Carboxyfluorescein tagged conjugates **geo106**. To a stirred solution of **geo107** (2 mg, 1.51 μmol) and **geo98** (0.7 mg, 1.51 μmol) in dry DMF (1.33 mL), under argon, was added DIPEA (2.62 μL, 15.1 μmol), and the mixture was stirred at room temperature for 48 h. The solvent was evaporated, and the product was purified by HPLC (method 5, Appendix A) to yield **geo106** (1.6 mg, 63%) as a yellow solid. **geo106**: ^1^H NMR (500 MHz, DMSO-*d*_6_) δ 9.10 (s, 1H), 8.83 (s, 1H), 8.72 (s, 1H), 8.63 (s, 1H), 8.45 (d, *J* = 12.9 Hz, 1H), 8.39 (s, 1H), 8.34 (d, *J* = 8.0 Hz, 1H), 8.29 (s, 1H), 8.23–8.15 (m, 2H), 8.13 (d, *J* = 8.1 Hz, 1H), 8.04–8.02 (m, 2H), 7.96 (s, 1H), 7.88 (s, 1H), 7.83 (s, 1H), 7.63 (t, *J* = 6.9 Hz, 2H), 7.35–7.30 (m, 1H), 7.19 (s, 2H), 6.96–6.90 (m, 2H), 6.86 (d, J = 8.2 Hz, 1H), 6.69–6.56 (m, 4H), 6.55–6.51 (m, 1H), 6.17 (d, *J* = 8.0 Hz, 1H), 6.16–6.04 (m, 1H), 4.92–4.79 (m, 26H), 4.69–4.63 (m, 1H), 4.60–4.53 (m, 1H), 4.49 (dd, *J* = 13.8, 7.3 Hz, 1H), 4.38–4.25 (m, 2H), 4.23–4.16 (m, 1H), 4.14 (d, *J* = 9.3 Hz, 1H), 4.12–4.04 (m, 2H), 3.63–3.54 (m, 4H), 3.52–3.46 (m, 4H), 3.17 (s, 1H), 3.14–3.10 (m, 2H), 3.09–3.03 (m, 2H), 2.94 (q, *J* = 15.0, 2H), 2.80–2.73 (s, 3H), 2.09 (d, *J* = 9.3 Hz, 1H), 2.04–1.96 (m, 2H), 1.80 (s, 6H), 1.59–1.51 (m, 3H), 1.50–1.43 (m, 3H), 1.40–1.36 (m, 2H), 1.31–1.22 (m, 7H), 1.17 (t, *J* = 7.4 Hz, 4H), 1.16–1.07 (m, 2H). ESI-MS, negative mode: *m/z* calcd mass for C_86_H_95_N_17_O_20_ [M/2-H]^−^ = 841.84, was found 841.60.

### 2.2. Chemostability Assays

#### 2.2.1. Stability in Buffer Solutions

The stability of the conjugates was examined by performing chemostability experiments at two different buffer solutions (pH = 5.2 and 7.4). c(RGDyK)-based conjugates were dissolved in 5 μL DMSO and transferred to 0.5 mL of the relevant buffer solution (acetate or phosphate aquatic buffer solutions). The mixture was then incubated at 37 °C, samples were collected at predetermined time points (0, 1, 2, 3, 4, 5, 24 and 48 h), and analyzed by LC-MS. Results are presented as the mean ± of SD after repeating the experiment three times.

#### 2.2.2. Stability in Dulbecco’s Modified Eagle Medium

Conjugates **geo75**, **geo77**, **geo85** and **geo107** (200 μg in 5 μL DMSO) were diluted in 0.5 mL DMEM (+10% fetal bovine serum) mixture and incubated at 37 °C. At predetermined time intervals, 50 μL of the mixture was removed and quenched with 150 μL water/acetonitrile/formic acid solution (100/100/0.1 volume ratio) and the sample was analyzed by LC-MS. Three independent experiments were carried out and the results are presented as the mean ± standard deviation.

#### 2.2.3. Stability in Human Plasma

First, 250 μg of (JH-VII-139-1)-c(RGDyK)-based conjugates were diluted in 5 μL DMSO and the mixture was added to 0.5 mL of human plasma. The mixtures were incubated at 37 °C. 50 μL of aliquot was removed at predetermined time points and quenched with 150 μL ice-cold acetonitrile (+0.1% formic acid). Then, the mixture was centrifuged at 10,000 rpm for 10 min. Then, 50 μL of supernatant was added to 50 μL of ultrapure water (+0.1% formic acid) and analyzed by LC-MS. Results are presented as the mean ± SD of three independent experiments.

### 2.3. In Vitro Kinase Assays

The pGEX-2T bacterial expression vector (Amersham Biosciences GmbH, Freiburg, Germany) was used to construct plasmids that encode human SRPK1 and a fragment of the *N*-terminal domain of turkey LBR comprising amino acids 62–92 (LBRNt(62–92)) [36]. The GST-fusion proteins were produced in bacteria and purified using glutathione-Sepharose (Amersham Biosciences) according to the manufacturer’s instructions. Kinase assays were carried out in a total volume of 25 μL containing 0.5 μg GST-SRPK1, 1.5 μg GST-LBRNt(62–92) as substrate, 12 mM Hepes pH 7.5, 10 mM MgCl_2_, 25 μM ATP, and increasing concentrations of the inhibitors as indicated. In all assays, the final concentration of dimethyl sulfoxide (DMSO) was adjusted to 4%. Phosphorylated GST-LBRNt(62–92) was detected by autoradiography using Super RX (Fuji medical X-ray film,(Fujifilm Holdings Corporation, Tokyo, Japan)). Incorporated radioactivity was quantified by excising the radioactive bands from the SDS-PAGE gel and scintillation counting.

### 2.4. Cell Toxicity

#### 2.4.1. Cell Culture

HeLa, MCF7, MDA-MB-231 and K562 cancer cell lines were provided by ATCC (American Type Culture Collection). Cells were incubated at 37 °C with 5% CO_2_. HeLa, MCF7 and MDA-MB-231 cells were maintained in DMEM, while K562 cells were maintained in RPMI medium, both supplemented with 10% (*v/v*) fetal bovine serum (FBS) and antibiotics/antimycotics.

#### 2.4.2. MTT Assays

HeLa, MCF7, MDA-MB-231 and K562 cells were seeded in 96-well plates (3 × 10^3^ cells per well) and grown in medium supplemented with 10% FBS as monolayers at 37 °C in a 5% CO_2_ incubator. One day after seeding, cells were exposed to increasing concentrations of the inhibitors for 48 h. The viability of the cells was estimated by an (3-(4,5-imethylthiazol-2-yl)-2,5-diphenyltetrazolium bromide (MTT) metabolic assay as described previously [37], and the percentage growth inhibition (GI_50_, TGI, IC_50_) was calculated according to National Cancer Institute recommendations as referred to Leonidis et al. [26]. Values shown represent the means ± SE of three independent experiments.

### 2.5. Fluorescence Microscopy

MCF7 and MDA-MB-231 cells were maintained in DMEM medium supplemented with 10% fetal bovine serum (FBS) at 37 °C in 5% CO_2_. Cells were seeded in 24-well plates with 5 × 10^4^ cells per well on coverslips, and after 24 h they were treated with JH-VII-139-1-c(RGDyK) hybrid compound, **Geo106** (5 µM), for 2, 24 and 48 h. To test the effect of okadaic acid (endocytosis inhibitor), cells were preincubated, prior to addition of **Geo106**, with okadaic acid (100 nM) for 30 min. After the incubation period, the cells coverslips were fixed with 4% paraformaldehyde in PBS for 10 min at room temperature. Following the quenching of paraformaldehyde (100 mM Tris-HCl pH 7.5), the cells were washed with PBS and DNA was stained with DAPI. The coverslips were mounted with mounting medium (0.01% *p*-phenylenediamine and 50% glycerol in PBS) and visualized with Zeiss LSM 780 confocal microscope, using the Zen 2011 software.

### 2.6. Zebrafish Angiogenesis Assay

#### 2.6.1. Zebrafish Screening Assays

The *Tg(kdrl:gfp)^s843^* [38] was used to measure angiogenesis. Zebrafish breeding was carried out according to the European Directive 2010/63 and the Recommended Guidelines for Zebrafish Husbandry Conditions [39].

Zebrafish *Tg(kdrl:gfp)^s843^* embryos were raised in E3 medium up to 24 hpf; then dechorinated and finally transferred to a 12-well plate (10 embryos per well). Each compound was dissolved in DMSO and then added in the wells at the following final concentrations: 10 μM JH-VII-139-1 and c(RGDyK), and 1.3 μM **geo75**, **geo77**, **geo85**, **geo107**. Treated embryos and control embryos (at the corresponding DMSO concentration) were kept at 28 °C and imaged at 72 hpf.

#### 2.6.2. Zebrafish Imaging Analysis

After anesthetizing embryos with 0.4% tricaine methanesulfonate (MS222), a fluorescent image was acquired using a Hamamatsu ORCA-Flash4.0LT digital camera. We used the total length of 8 intersegmental vessels (ISVs) per embryo and the width from each of the above ISVs was measured at three different points, i.e., the top, the middle, and the bottom point of each ISV, and then the average width was estimated. We measured the 8 ISVs that flank the end of the yolk extension. Length and widths were measured using the open-source software ImageJ 1.53a (http://imagej.nih.gov/ij (accessed on 30 May 2022)). A total of 30 embryos (4–7 per group) from independent experiments were measured. Statistically significant differences in total width, or in the length of ISVs were calculated using ordinary one-way ANOVA between the average lengths or widths for each embryo in GraphPad Prism 9.

## 3. Results

### 3.1. Synthesis of Compounds

#### 3.1.1. Optimized Synthesis of SRPK1 Inhibitor, JH-VII-139-1

The synthesis of the Alectinib core structure that eventually leads to JH-VII-139-1 is presented in Figure 1. Most Alectinib analogues, including JH-VII-139-1 are fused polycyclic indoles that combine the indole moiety with a polysubstituted tetralone. To synthesize the polycyclic scaffold **6a**, Fischer indole synthesis was implemented based on Kinoshita et al. [30]. Initially, 7-methoxy-2-tetralone **1** was emethylated with MeI to produce **2**, which was brominated with NBS to give the desired tetralone **3**. Then, 3-aminobenzonitrile was converted to 3-hydrazinylbenzonitrile **5**, which reacted with **3** in a Fischer indole synthesis reaction. After removing the unwanted regioisomer **6b** by filtration, **6a** was isolated by column chromatography in satisfying yield (50%) compared to similar indole synthesis of the literature [30]. Oxidation of **6a** with DDQ afforded **7**, which was subsequently alkylated with Et_3_B, Pd(dppf)Cl_2_ and Cs_2_CO_3_ in a Suzuki–Miyaura reaction, resulting in **8**. To modify the scaffold and insert the required pyrazole moiety that will lead to the synthesis of JH-VII-139-1, Zeisel–Prey ether cleavage with pyridinum chloride at high temperatures was performed to form phenol **9**. Finally, **9** reacted with *N*-phenyl-bis(trifluoromethanesulfonimide) to produce the triflate **10**, which was subsequently coupled to 1*H*-pyrazole-3-boronic acid **13** (Figure 2).

The synthesis of 1*H*-pyrazole-3-boronic acid **13,** as well as the synthesis of JH-VII-139-1, are outlined in Figure 2. Pyrazole was protected with a DHP group leading to the formation of **12**, which was then selectively borylated with (MeO)_3_B and *n*-BuLi [29,30]. The DHP-protected boronic acid was then deprotected under acidic conditions to give **13**, which reacted with triflate **10**. The handling of **10** proved to be difficult compared to the iodide analogue used in the JH-VII-139-1 synthesis proposed by Gray et al. [27], since it was easily hydrolyzed in H_2_O and heat. To avoid the hydrolysis of triflate **10**, dioxane was used as the sole solvent, which unfortunately led to no reaction. The presence of H_2_O in the reaction proved to be crucial; therefore, the reaction was examined at different temperatures. The reaction was also tested under different conditions and solvents, where in general the starting material **10** was consumed, leading to the hydrolyzed byproduct **9** and the byproduct **14**, as depicted in Table 1. After several attempts, microwave irradiation improved the yield of JH-VII-139-1 to 80% and reduced the formation of **9** to 12%. It should be noted that the yield 80% for JH-VII-139-1 is significantly improved compared to the 48% yield reported by Gray et al. using a similar methodology [27].

#### 3.1.2. Synthesis of (JH-VII-139-1)-c(RGDyK) Peptide–Drug Conjugates

Our first efforts focused on the conjugation of JH-VII-139-1 with c(RGDyK) through a double carbamate linker. The pentapeptide c(RGDyK) was prepared using solid-phase peptide synthesis (SPPS) and head-to-tail cyclization based on the methodology of Davis et al. followed by semipreparative HPLC purification [40]. The synthesis of **geo75** conjugate is summarized in Figure 3.

Starting from triethylene glycol **15**, the highly reactive dinitrophenyl carbonate **16** was obtained after reaction with 4-nitrophenyl chloroformate, Et_3_N in DCM (Figure 3). Then, dropwise addition of JH-VII-139-1 solution was performed to a solution of **16**, TEA and DMF to give the monosubstituted **17** in 55% yield. Careful handling and low concentrations of reactants reduced the unwanted double nucleophilic acyl substitution of **16** by JH-VII-139-1. Finally, **17** was coupled to the ε-amino group of lysine on c(RGDyK) under basic DIPEA conditions to afford conjugate **geo75** in 56% yield. Conjugate **geo75** was purified by semipreparative reversed-phase HPLC and its purity was examined by LC-MS, and it was found greater than 98%.

After achieving conjugation of JH-VII-139-1 and c(RGDyK) through a dicarbamate linker, linkers with different length or greater chemical stability were used for the preparation of new derivatives such as the diamide conjugate **geo77** (Figure 4). The synthesis started with the commercially available glutaric acid **18** which reacted with NHS and EDCi to quantitatively form the activated NHS glutarate **19**, which then reacted with JH-VII-139-1 in basic conditions and afforded the JH-VII-139-1 derivative in 47% yield. The subsequent reaction of **20** with oligopeptide c(RGDyK) and DIPEA eventually produced conjugate **geo77** in 45% yield, which was purified by HPLC with purity higher than 98%.

In an attempt to increase the chemical stability, triethylene glycol in **geo75** synthesis was replaced by 2,2′-(ethylenedioxy)bis(ethylamine), which could lead to a bis-urea linker. The synthetic procedure is outlined in Figure 5. In detail, highly reactive dicarbamate **22** was formed in a similar fashion by reacting 2,2′-(ethylenedioxy)bis(ethylamine) with 4-nitrophenyl chloroformate and EDCi. Then, JH-VII-139-1 reacted with **22** in basic conditions, and after careful handling, **23** was obtained in good yield (66%). Compound **23** was then coupled to c(RGDyK) by forming a second urea moiety, providing **geo85** in 88% yield. Conjugate **geo85** was easier to handle, and was isolated in greater overall yield compared to **geo75**, as urea derivatives were more stable and did not hydrolyze as readily as their carbamate counterparts. Conjugate **geo85** was isolated after semipreparative HPLC with purity higher than 97%.

Afterwards, a linker that would increase the distance between JH-VII-139-1 and c(RGDyK) and could carry another bioactive moiety was employed. Lysine was a good candidate, as it could extend the triethylene glycol chain and bear both a carboxylic and an amino group. This approach that eventually led to the synthesis of conjugate **geo107** is highlighted in Figure 6. The synthesis starts with a nucleophilic attack by *N*_ε_-Boc-*L*-lysine to the highly reactive 4-nitrophenyl carbamate of **23** that produces Boc-protected **24** in 63% yield. The carboxyl group is then subjected to in situ NHS activation and direct amidation with c(RGDyK) oligopeptide and DIPEA to form Boc-protected **25** in 30% yield over two steps. The final conjugate **geo107** was obtained after a rapid Boc-deprotection with TFA and triethylsilane (TESH) followed by semi-preparative reversed-phase HPLC purification (purity of **geo107** > 97%).

#### 3.1.3. Synthesis of Fluorescein-Tagged (JH-VII-139-1)-c(RGDyK) Conjugate

To gain insight into the cellular entry mechanism and the cellular biodistribution of (JH-VII-139-1)-c(RGDyK) conjugates, fluorescence labeling of **geo107** with fluorescein was undertaken (Figure 7). Two molecules of rescorcin reacted with trimellitic anhydride to produce a mixture of 5- and 6-carboxy-fluoresceins **28** in quantitative conversion [41]. Activation of 5(6) carboxyl group with NHS and DIPEA led to the formation of NHS ester **29** in 48% yield [42], which subsequently reacted with **geo107** under basic conditions and formed the 5(6)-carboxyfluorescein-tagged conjugate **geo106** in 63% yield. The crude photosensitive material was then subjected to semi-preparative HPLC and **geo106** was isolated with 96% purity.

### 3.2. In Vitro Stability Studies

The stability of JH-VII-139-1 and its c(RGDyK) conjugates **geo75**, **geo77**, **geo85** and **geo107** was studied at pH = 5.2 and 7.4, in cell medium and human plasma. All experiments were carried out at 37 °C, and samples were analyzed with LC-ESI-MS at predetermined time intervals. Starting with JH-VII-139-1, the compound was found to be very stable under all conditions tested as presented in Figure 2 (Appendix A). The c(RGDyK) conjugates exhibit different stability profiles as analyzed below. Hydrolysis and JH-VII-139-1 release were the main reactions, with the half-lives of the conjugates ranging from minutes to hours.

#### 3.2.1. Chemostability Profile of Conjugate **geo75**

Dicarbamate **geo75** was tested for its stability to buffer solutions and it was completely stable at pH = 5.2 for more than 48 h, while after 24 h its residual concentration at pH = 7.4 retained more than 51% of its initial concentration. In addition, the conjugate had a declining stability pattern in Dulbecco’s modified eagle medium (DMEM) and human plasma, where it was almost completely hydrolyzed after 9- and 1-h respectively, exhibiting half-lives t_1/2_ = 2 h 36 min and t_1/2_ = 11 min (Figure 3, Appendix A).

#### 3.2.2. Chemostability Profile of Conjugate **geo77**

The stability of the conjugate **geo77** was also examined in the same conditions (Figure 4, Appendix A). In detail, **geo77** was less stable at pH = 5.2 buffer, having a half-life t_1/2_ = 14 h 42 min compared to the t_1/2_ > 48 h of **geo75.** At neutral pH and in DMEM solutions, the conjugate was readily hydrolyzed leading to half-lives t_1/2_ = 1 h 44 min and t_1/2_ = 0 h 50 min, respectively. In human plasma, the conjugate was also unstable, and the half-life time was less than ten minutes.

#### 3.2.3. Chemostability Profile of Conjugate **geo85**

Since both carbamate and amide conjugates, **geo75** and **geo77** exhibited short half-lives during their incubation in cell media and human plasma, our efforts were focused on the urea derivative **geo85** (Figure 5, Appendix A). Conjugate **geo85** proved to be completely stable in both buffer media having half-lives that exceeded 48 h. **Geo85** maintained its stability in DMEM and human plasma, although a 10% hydrolysis was observed after 24 h. Therefore, **geo85** was the most stable conjugate and proved to be an excellent candidate for fluorescent tagging and mechanistic studies.

#### 3.2.4. Chemostability Profile of **geo107**

The stability profile of **geo107** was analogous to **geo85**, as it maintained half-lives t_1/2_ > 48 h in all stability experiments (Figure 6, Appendix A). Hydrolysis and JH-VII-139-1 was also detected. Residual concentrations of the conjugate dropped no further than 80% of its initial concentration even in cell media or human plasma.

### 3.3. Biological Activity of the Synthesized Compounds

#### 3.3.1. Inhibition of Kinase Activity by JH-VII-139-1-c(RGDyK) Conjugates

The inhibitory activity of JH-VII-139-1 and RGD-conjugated compounds was evaluated by in vitro kinase assay. As shown in Table 2, both JH-VII-139-1 and the conjugated compounds had a significant effect on SRPK1 activity, with IC_50_ values at the nM level (Appendix A). Therefore, the conjugation of JH-VII-139-1 to the peptide did not affect the inhibitory activity against SRPK1.

#### 3.3.2. Cytotoxicity of JH-VII-139-1-c(RGDyK) Conjugates

The cytotoxic activity of JH-VII-139-1 and JH-VII-139-1-c(RGDyK) hybrid compounds was evaluated at different concentrations (0.5–50 μM) over a panel of cell lines including HeLa cervical cancer, MCF7 mammary carcinoma, the triple-negative breast cancer MDA-MB-231 and K562 lymphoblast cells. Integrin receptors are highly overexpressed on the surface of many types of cancer [43]. The metastatic breast cancer cell lines MDA-MB-435 [44,45] and MCF-7 [46,47], as well as HeLa cells [48], express high levels of α_V_β_3_ integrins. On the other hand, K562 cells express very low levels of α_V_β_3_ integrins [22].

The cytotoxic and cytostatic activities of the JH-VII-139-1-c(RGDyK) hybrid compounds were estimated by three concentration-dependent parameters: GI_50_ (concentration that results in 50% growth inhibition), TGI (concentration that results in total growth inhibition or cytostatic effect), and IC_50_ (concentration that results in 50% growth cytotoxic effect) (Table 3). JH-VII-139-1 and the hybrid compounds, **geo75** and **geo77**, showed a significant cytostatic (GI_50_ = 4–12 μM) and cytotoxic (IC_50_ = 3–18 μM) effect against HeLa, MCF-7, MDA-MB-231 and K562 cancer cells. The most active compound was **geo77**, showing IC_50_ and GI_50_ values of 3 μM and 4 μM, respectively in the most sensitive cell line MCF-7. **Geo85** exhibited a slightly cytostatic effect (GI_50_ = 33 μM), with a minor cytotoxicity (IC_50_ = 45 μM) only against MCF-7 cells.

#### 3.3.3. Cellular Uptake of c(RGDyK) Conjugate **geo106**

The cellular uptake of **geo106**, a c(RGDyK) conjugate carrying a fluorescent label, was examined by fluorescent microscopy in MCF7 and MDA-MB-231 cancer cell lines (Figure 7). Confocal image data showed efficient cellular uptake of **geo106** even after a short time of incubation (2 h) in MDA-MB-231 cells, with the most intense fluorescent signal obtained in 48 h in both cancer cell lines.

To demonstrate the role of the RGD moiety in targeting the conjugate to a_v_β_3_ integrin-rich tumor cells, MDA-MB-231 cells were preincubated with okadaic acid (an inhibitor of endocytosis) prior to addition of **geo106** [49]. Confocal image data showed a decreased fluorescence signal, confirming that the cellular uptake of the conjugates was mediated by endocytosis (Figure 8).

#### 3.3.4. In Vivo Zebrafish Angiogenesis Studies

To determine if compounds c(RGDyK), JH-VII-139-1, **geo75**, **geo77**, **geo85**, and **geo107** affect angiogenesis, we used the transgenic zebrafish line *Tg(kdrl:gfp)*. The compounds were added in the embryo medium at 24 h post fertilization (hpf) and angiogenesis was monitored by imaging the intersegmental vessels (ISVs) at 72 hpf. The length as well as the width of the ISVs were used as measurements to quantify angiogenesis.

Our results show that the conjugates significantly inhibit the width of ISVs (Figure 9), while exerting a milder effect on their length (Figure 10). More specifically, **geo75**, **geo77**, **geo85**, and **geo107** significantly inhibited width angiogenesis at concentrations of 1.3 μM, while JH-VII-139-1 at 10 μΜ and c(RGDyK) showed a non-significant trend. On the other hand, **geo77** also exhibited an effect on ISV length inhibition, while the other compounds showed a non-significant trend at the concentrations tested. In this study, no other additional morphological phenotypic changes were observed.

## 4. Discussion

In the present study, we disclose the synthesis and biological evaluation of several JH-VII-139-1-c(RGDyK) conjugates, which inhibit SRPK1 kinase. To develop novel antiangiogenic compounds JH-VII-139-1, a potent SRPK1 inhibitor, was coupled to c(RGDyK) peptide using different linkers. Four c(RGDyK) conjugates **geo75**, **geo77**, **geo85** and **geo107** were synthesized, which were examined for their stability at pH = 5.2 and 7.4 buffer solutions, cell medium and human plasma. Conjugates **geo75** and **geo77** demonstrated limited stability and were rapidly hydrolyzed releasing JH-VII-139-1. In human plasma, the same conjugates had short half-lives of 10–50 min. On the other hand, **geo85** and **geo107** showed lasting stability in buffer solutions, cell medium and human plasma. In vitro kinase assay against SRPK1 showed that all the tested JH-VII-139-1-c(RGDyK) conjugates retained the inhibitory activity against SRPK1 in the nanomolar range (30–42 nM). The cytotoxic activity of the synthesized compounds was tested against the cancer cell lines HeLa, MCF7, MDA-MB-231 and K562. Among them, MCF7 proved to be the most sensitive (lower IC_50_ values) to JH-VII-139-1 and its conjugates. Overall, the conjugates **geo75** and **geo77** bearing the cleavable linkers slightly improved the antiproliferative activity compared to JH-VII-139-1 against integrin α_v_β_3_ overexpressing cells (HeLa, MDA-MB-231, MCF7). However, the observed activities are in part attributed to an early JH-VII-139-1 release as the stability studies indicate. The stable conjugate **geo85** induced selective cytotoxicity towards MCF7 cancer cells in the mid-micromolar range, whereas it was significantly less active against HeLa and MDA-MB-231 cells. The cellular uptake of the conjugate **geo106**, which carries a fluorescent label, was examined by fluorescent microscopy in MCF7 and MDA-MB-231 cancer cell lines. Confocal image data showed efficient cellular uptake of **geo106** even after a short time probably by a mechanism of integrin mediated endocytosis. Then, the effect of (JH-VII-139-1)-c(RGDyK) conjugates on angiogenesis was examined in vivo in zebrafish embryos. Notably, the new conjugates significantly inhibit zebrafish width angiogenesis and exert a milder effect in length angiogenesis. Among the tested conjugates, the stable **geo85** was the most potent inhibitor of zebrafish width angiogenesis.

## 5. Conclusions

The synthesized (JH-VII-139-1)-c(RGDyK) conjugates retained the inhibitory activity of JH-VII-139-1 against SRPK1, while exhibiting different cytotoxicity profile against cancer cells expressing integrins to various extend. The activities were strongly related to the stability of the linkers and the early release of JH-VII-139-1. The stable conjugate **geo85** induced increased targeting ability against MCF7 cancer cells; however it was mildly cytotoxic with GI_50_ = 33 ± 0.9 μΜ, TGI = 64 ± 0.9 μΜ, IC_50_ = 45 ± 1.0 μΜ. Remarkably, all (JH-VII-139-1)-c(RGDyK) conjugates displayed in vivo antiangiogenic effects, although a more significant effect was observed for **geo85** against width angiogenesis in zebrafish embryos. Since all the synthesized conjugates inhibit SRPK1 with similar potency, the observed difference in biological activities related to cytotoxicity and anti-angiogenesis, depend essentially on the different drug release and endosomal escape mechanisms, as well as the stability of the conjugates. In addition, a_v_β_3_ integrin and SRPK1 signaling may defer considerably in different cancer cell lines. Further experiments, in particular in vivo, are clearly necessary to reveal the full potential of such peptide conjugates in cancer therapy and angiogenesis-related diseases.

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
