# Peer review of "Synthesis and Anti-Angiogenic Activity of Novel c(RGDyK) Peptide-Based JH-VII-139-1 Conjugates"

_pharmaceutics, 2023, doi:10.3390/pharmaceutics15020381_

Round 1
Reviewer 1 Report
The manuscript by Sarli and co-workers is focused on the chemical synthesis and biological evaluation of several novel covalent conjugates bearing the popular alphavbeta3 integrin binder c(RGDyK) and the SRPK1 inhibitor JH-VII-139-1.
The search for new dual molecules namely, peptide drug-conjugates, where a receptor-selective peptide targeting unit is linked to an unselective cytotoxic agent is a flourishing field of contemporary research, which intends to merge the competence of the constituting units while conferring selectivity of action and minimization of off-target toxicity. In this regard, the work is interesting and merits consideration for publication.
The central idea of the work is clearly presented, and the experimental results are reported in detail and precision. Of particular interest is the evaluation of the stability of the new hybrid molecules at various pH values, and different media (e.g. human plasma, DMEM, …). These results clearly indicated that bis-carbamate and bis-amide conjugates geo75 and geo77 were highly unstable and easily liberated the free drug, while the bis-urea conjugates geo85 and geo107 were highly stable.
This behavior heavily impacted the biological profile of the conjugates, with the first two conjugates almost overlapping the behavior of the free drug (e.g. same IC50 for SRPK1, almost the same cytostatic and cytotoxic results), and geo85 being a new, chemically stable molecule with its own (and interesting) biological properties.
To be useful, though, this work should be complemented with additional information, and major revision is accordingly required.
11) The new conjugates geo75/geo77/geo85 should be assayed in vitro in order to confirm their affinity toward the alphaVbeta3 integrin receptor. It is highly probable that they maintain this affinity, but it should be experimentally proven, since the attached cargo could somehow impact on this data.
22) The expression of the alphavbeta3 integrin in the cell lines used in this work (and possibly of alphaVbeta5 integrin which is also relevant to angiogenesis) should be experimentally evaluated. It is not sufficient to report the expression of these lines in the literature, since they could vary. This information is crucial in order for considering a possible correlation between the percentage of the integrin expression and the biological results.
33) From the uptake experiments with geo106, a fluorescent analogue of geo85, it seems that this conjugate easily enters both the MCF7 and MDA-MB-231 cells; however, geo85 shows cytotoxicity only against MCF7: how do the authors interpret this result? Could this result depend upon the different in-cell concentration of the conjugate in the two cell lines?
44) The in vivo experiments should have considered also the combined administration of the free drug and the RGD ligand (at 1.3 micromolar), since possible synergistic effect could be observed. This could explain the behavior observed for the unstable conjugates geo75 and geo77, which show an increased antiangiogenic effect compared to the free drug, even if it is highly probable that they are completely dissociated into the RGD and drug components under the experimental conditions.
55) The authors proved that the geo107 conjugate entered cells via endocytosis (via okadaic acid pre-treatment). However, they did not prove that this endocytosis is alphavbeta3 integrin mediated. To evidence the active role exerted by this integrin, the use of a specific alphavbeta3 antibody or competitive experiments should be carried out.
66) The discussion section should contain comments on the results of the above requested experiments.
Minor points:
A series of minor points is directly reported in the attached revised manuscript file.
Overall, this reviewer thinks that this work may be published in Molecular Pharmaceutics, provided that the above points are carefully addressed.

Author Response
11) The new conjugates geo75/geo77/geo85 should be assayed in vitro in order to confirm their affinity toward the alphaVbeta3 integrin receptor. It is highly probable that they maintain this affinity, but it should be experimentally proven, since the attached cargo could somehow impact on this data.
Our answer:We appreciate the reviewer's suggestion. However, there are many examples in the literature referring to the use of RGD peptides (cyclized and linear) in drug delivery. Conjugation at lysine of cRGDyK peptide is considered as a safe option to keep the affinity toward the alphaVbeta3 integrin receptor.[i], [ii]
22) The expression of the alphavbeta3 integrin in the cell lines used in this work (and possibly of alphaVbeta5 integrin which is also relevant to angiogenesis) should be experimentally evaluated. It is not sufficient to report the expression of these lines in the literature, since they could vary. This information is crucial in order for considering a possible correlation between the percentage of the integrin expression and the biological results.
Our answer: It is true that integrin levels vary in different cell lines, however our group in 2022 reported the low levels of alphavbeta3 integrins in K562 cancer cells.2 In another work in 2021, we have also shown that MCF7 and MDA-MB231 express high and moderate levels of alphavbeta3 integrins.[iii] This is the main reason that we did not repeat the experiments.
33) From the uptake experiments with geo106, a fluorescent analogue of geo85, it seems that this conjugate easily enters both the MCF7 and MDA-MB-231 cells; however, geo85 shows cytotoxicity only against MCF7: how do the authors interpret this result? Could this result depend upon the different in-cell concentration of the conjugate in the two cell lines?
Our answer: The conjugate with the fluorescent tag geo107 easily enters MCF7 and MDA-MB-231 cells. The cytotoxicity could be a result of a distinct role of the SRPK1 kinase or different endosomal escape mechanisms in different cancer cells. This statement was added in the Conclusion section.
44) The in vivo experiments should have considered also the combined administration of the free drug and the RGD ligand (at 1.3 micromolar), since possible synergistic effect could be observed. This could explain the behavior observed for the unstable conjugates geo75 and geo77, which show an increased antiangiogenic effect compared to the free drug, even if it is highly probable that they are completely dissociated into the RGD and drug components under the experimental conditions.
Our answer: For the stable geo85 the anti-angiogenesis activities are due to the activity of the conjugate. The anti-angiogenesis properties of the conjugates are better than those of cRGDyK and JΗ-VII-139-1, since cRGDyK and JΗ-VII-139-1 have been administered in 10μΜ concentration. For the unstable derivatives, the activities derived from the conjugates and the parent compounds c(RGDyK) peptide and JΗ-VII-139-1. The synergistic effect of the parent drugs has not been studied, but it is possible from the observed findings.
55) The authors proved that the geo107 conjugate entered cells via endocytosis (via okadaic acid pre-treatment). However, they did not prove that this endocytosis is alphavbeta3 integrin mediated. To evidence the active role exerted by this integrin, the use of a specific alphavbeta3 antibody or competitive experiments should be carried out.
Our answer: Based on the structure of the molecules and our previous research on cRGDyk conjugates, the most probable mechanism is an integrin mediated endocytosis.2 Since no experiments were performed for the determination of the endocytosis mechanism, the sentence line849 in the discussion section was rephrased.
66) The discussion section should contain comments on the results of the above requested experiments.
Our answer: Comments have been added in Conclusions section.
Minor points: A series of minor points is directly reported in the attached revised manuscript file.
Our answer: This has been addressed in the current manuscript.
[i] Chatzisideri T, Leonidis G, Karampelas T, Skavatsou E, Velentza-Almpani A, Bianchini F, Tamvakopoulos C, Sarli V. Integrin-Mediated Targeted Cancer Therapy Using c(RGDyK)-Based Conjugates of Gemcitabine. J Med Chem. 2022 Jan 13;65(1):271-284. doi: 10.1021/acs.jmedchem.1c01468. Epub 2021 Dec 30. PMID: 34967607.
[ii] Katsamakas S, Chatzisideri T, Thysiadis S, Sarli V. RGD-mediated delivery of small-molecule drugs. Future Med Chem. 2017 Apr;9(6):579-604. doi: 10.4155/fmc-2017-0008. Epub 2017 Apr 10. PMID: 28394627.
[iii] Chatzisideri T, Dalezis P, Leonidis G, Bousis S, Trafalis D, Bianchini F, Sarli V. Synthesis and biological studies of c(RGDyK) conjugates of cucurbitacins. Future Med Chem. 2021 May;13(10):877-895. doi: 10.4155/fmc-2020-0309. Epub 2021 Apr 16. PMID: 33858195.
Reviewer 2 Report
The manuscript of Sarli e al. deals with the synthesis of new peptide-based JH-VII-139-1 conjugates whose cytotoxic and antiangiogenic activity was also investigated. New and interesting data are offered to the reader, however some revisions are required before its publication.
Major concerns:
In the biological part is not clear why authors did not investigate the kinase inhibitory activity and the cytotoxicity of compound geo107 that was studied by in vivo assays; please justify.
Despite its kinase activity, compound geo85 presents a different behavior compared to the other conjugates. Authors should try to explain the reason.
Minor concern:
All synthetic schemes should be revised; yields and conditions are not in accordance to those reported in the experimental part.
Author Response
In the biological part is not clear why authors did not investigate the kinase inhibitory activity and the cytotoxicity of compound geo107 that was studied by in vivo assays; please justify.
Our answer: Compound geo107 has a high similarity with geo85. The difference in the structure is far from the binding point to kinase.
Despite its kinase activity, compound geo85 presents a different behavior compared to the other conjugates. Authors should try to explain the reason.
Our answer: Comments have been added in Conclusions section.
Minor concern:
All synthetic schemes should be revised; yields and conditions are not in accordance to those reported in the experimental part.
Our answer: This has been addressed in the current manuscript.
Round 2
Reviewer 1 Report
Corrected sentence in page 25, line 894 is not clear. Please, correct.
Author Response
This has been addressed in the current manuscript.
